# ELEVATER: A Benchmark and Toolkit for Evaluating Language-Augmented Visual Models

**Chunyuan Li**[*1♠]**, Haotian Liu**[*2]**, Liunian Harold Li**[3]**, Pengchuan Zhang**[1]**, Jyoti Aneja**[1]
**Jianwei Yang**[1]**, Ping Jin**[1]**, Houdong Hu**[1]**, Zicheng Liu**[1]**, Yong Jae Lee**[2]**, Jianfeng Gao**[1]
[1]Microsoft      [2]University of Wisconsin–Madison      [3]UCLA

## Abstract

Learning visual representations from natural language supervision has recently shown great promise in a number of pioneering works. In general, these language-augmented visual models demonstrate strong transferability to a variety of datasets and tasks. However, it remains challenging to evaluate the transferablity of these models due to the lack of easy-to-use evaluation toolkits and public benchmarks. To tackle this, we build ELEVATER [1] , the first benchmark and toolkit for evaluating (pre-trained) language-augmented visual models. ELEVATER is composed of three components. $(i)$ Datasets. As downstream evaluation suites, it consists of 20 image classification datasets and 35 object detection datasets, each of which is augmented with external knowledge. $(ii)$ Toolkit. An automatic hyper-parameter tuning toolkit is developed to facilitate model evaluation on downstream tasks. $(iii)$ Metrics. A variety of evaluation metrics are used to measure sample-efficiency (zero-shot and few-shot) and parameter-efficiency (linear probing and full model fine-tuning). ELEVATER is platform for *Computer Vision in the Wild (CVinW)*, and is publicly released at `https://computer-vision-in-the-wild.github.io/ELEVATER/`.

## 1 Introduction

Visual recognition has become ubiquitous in our society [76], with applications in geo-localization [66], action recognition [73], street number transcription [58], satellite remote sensing [31], medical imaging [80], self-driving cars [26], *etc.* Core to these applications are visual recognition tasks such as image classification (IC) and object detection (OD). It is of high value to develop transferable visual models that perform well on a wide range of downstream applications. By leveraging large web crawled image-text corpora, recent advances in language-augmented visual models such as CLIP [66] and ALIGN [35] have demonstrated strong transfer performance, making this direction one of the most practical visual learning approaches. The reason is twofold: $(i)$ open-set recognition is made possible by reformulating classification tasks as retrieval; $(ii)$ model generalization is improved as language supervision significantly increases the coverage of visual concepts for model training.

The success has immediately inspired many studies of large-scale model pre-training [88, 92, 89, 49, 57, 27, 47, 95]. However, these studies use their own evaluation settings based on customized sets of downstream tasks where the detailed process of adapting the models to these tasks is typically not accessible to the public. Thus, it is extremely difficult for researchers to fairly compare models and develop new models based on other people's works. To fill this gap, we develop an open-source benchmark and toolkit, ELEVATER, to make the research results (*e.g.,* model's task-level transferability) more rigorous, and reproducible. ELEVATER is composed of three components.

---

[*]Equal Technical Contribution   ♠Project Lead
[1]**E**valuation of **L**anguage-augm**e**nted **V**isual T**a**sk-level **T**ransf**er**

- **Benchmark (Datasets and Knowledge).** We build the first publicly available benchmark to evaluate the *large-scale task-level transferability* of language-augmented visual models. The benchmark consists of two challenges: *Image Classification in the Wild (ICinW)* with 20 IC datasets and *Object Detection in the Wild (ODinW)* with 35 OD datasets. A collected external knowledge base for each dataset which could be used for language data augmentation.

- **Comprehensive Metrics.** To measure the cost of deploying models for real-world applications, we measure a model's sample-efficiency in the zero-shot, few-shot and full-shot settings and parameter-efficiency in the linear probing and full model fine-tuning settings.

- **Reproducible Toolkit & Language-augmented Adaptation Methods.** We develop an open-source software toolkit to support model adaptation and evaluation. Automatic hyper-parameter tuning is employed to avoid human-in-the-loop tuning, thus reducing human labor and ensuring a fair comparison among different model checkpoints. We also present a set of new model adaptation methods for pre-trained language-augmented visual models. Our methods significantly outperform the traditional vision-only adaptation methods. These methods serve as baselines for the development of more advanced adaptation methods.

In addition, our empirical study leads to interesting findings. ($i$) Leveraging both text and vision in these models consistently yields better performance than vision-only in few-shot settings; In contrast, random initialization of the linear head in language-augmented visual models is sub-optimal. We also find that few-shot results are always better than zero-shot results, which is different from the results reported in [66]. ($ii$) For language-augmented visual models, linear probing performs better than full model fine-tuning in the few-shot settings. As the task-specific training data increases, fine-tuning outperforms linear probing. ($iii$) Our study shows that the use of external knowledge, including *explicit* knowledge of human-compiled thesaurus/dictionaries/documents and *implicit* knowledge stored in GPT3 [6], can improve zero-shot and few-shot learning performance. We summarize the pipeline to use ELEVATER in Figure 1, and organize the paper to focus on the benchmark and toolkit.

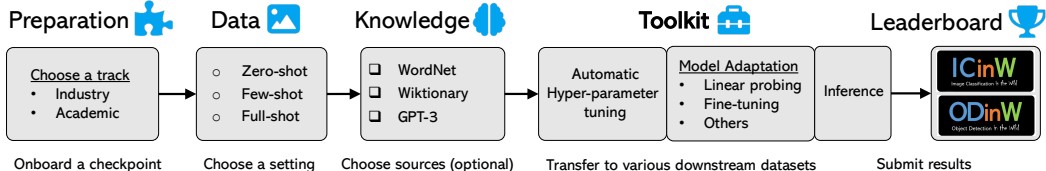

Figure 1: The illustrative pipeline to use ELEVATER to evaluate a model checkpoint.

## 2 Related Work: From Class-level to Task-level Transfer

Zero-shot learning in computer vision has been studied for decades. The research topic has witnessed two distinctive notions of zero-shot: the traditional *class-level zero-shot* that usually refers to the study of generalizing to unseen object categories [42], and the recently popular *task-level zero-shot* that refers to the study of generalizing to unseen datasets/tasks [44, 66]. In Table 1, we compare our benchmark against existing benchmarks. Existing zero-shot learning benchmarks are developed for the class-level zero-shot setting. They are usually in a single domain, with a manual split of categories to produce disjoint training and test categories, *e.g.,* Animal with Attributes (AwA) [43], Caltech-UCSD Birds-200 (CUB) [82], SUN [61], aPY [21], and ZS-ImageNet [67, 24]. For the OD problem, COCO [50] and LVIS [29] represent two well established datasets to compare various OD methods in a single domain, while UODB [86] is a multi-domain OD benchmark.

In contrast, our benchmark focuses on task-level transfer across domains, *i.e.,* it aims to evaluate the transferability of models, by pre-training from their own large corpus, then evaluating zero-shot performance on a diverse set of downstream datasets. This setting has been recently studied [44, 66, 45, 92], and is arguably more practical for real-world applications, as it brings the convenience towards the spirit of one-model-for-all. The well-known ImageNet-1K dataset [14] was originally proposed as a large dataset for model training and testing. It has also recently been considered as one downstream task to study zero-shot transfer [66, 45, 92]. Our work presents the first *public benchmark* to standardize the zero-shot task-level transfer setting. Note that visual task transfer has been previously explored in VTAB [94], which measures good visual representations as those that adapt to diverse, unseen tasks with an emphasis on few training examples. The pre-trained models and task adaptation in VTAB are considered for vision backbones only, and no language model/modality is involved. Our benchmark shares a similar spirit of task-level transfer to VTAB, but

| Problem | | **Benchmark Statistics** | | | | **Evaluation** | | |
| --- | --- | --- | --- | --- | --- | --- | --- | --- |
| | | #Datasets | #Image | #Concepts | Knowledge Source | Zero | Few | Full |
| Image Classification (IC) | AwA [43] | 1 | 30337 / 6985 | 40 / 10 | Attributes | ✓ | | |
| | CUB [82] | 1 | 8855 / 2933 | 150 / 50 | Attributes | ✓ | | |
| | SUN [61] | 1 | 12900 / 1440 | 645 / 72 | Attributes | ✓ | | |
| | aPY [21] | 1 | 12695 / 2644 | 20 / 12 | Attributes | ✓ | | |
| | ZS-ImageNet [67] | 1 | 1.2M / 54K | 1K / 360 | WordNet | ✓ | | |
| | ImageNet-1K [14] | 1 | 1.2M / 50K | 1K | WordNet | ✓ | | ✓ |
| | VTAB [94] | 19 | 2.2M / - | 940 | - | | ✓ | ✓ |
| | ELEVATER (ICinW) | 20 | 638K / 193K | 1151$^\diamond$ | WordNet, Wiki, GPT-3 | ✓ | ✓ | ✓ |
| Object Detection (OD) | COCO [50] | 1 | 83K / 41K | 80 | - | | | ✓ |
| | LVIS [29] | 1 | 120K / 40K | 1723 | WordNet | | | ✓ |
| | UODB [86] | 11 | 113K / 40K | 109 | - | | | ✓ |
| | ELEVATER (ODinW) | 35 | 132K / 20K | 314$^\diamond$ | WordNet, Wiki, GPT-3 | ✓ | ✓ | ✓ |

Table 1: Comparison of dataset statistics and evaluation settings. For existing zero-shot datasets in IC, the number of images and concepts are reported for development / evaluation stages separately. $^\diamond$ represents the total number of concepts in the benchmark to evaluate task-level transfer, and there is no train-evaluation category split as in class-level transfer.

strives to analyze the vital role of language and knowledge in visual transfer. All of them are usually evaluated in full-shot settings, without considering task-level transfer. We have further made several novel contributions to consolidate the benchmark: $(i)$ We add external knowledge for each dataset to cultivate new research directions in knowledge-augmented visual models, inspired by the success of knowledge in traditional class-level transfer. $(ii)$ We consider the full spectrum in measuring the sample-efficiency of task adaptation, including zero-shot, few-shot, and full-shot.

In this paper, we develop ELEVATER as a platform for "**computer vision in the wild**", whose ultimate goal is to develop a transferable foundation model/system that can *effortlessly* adapt to *a large range of visual tasks in the wild*. It consists of two key factors: $(i)$ **The task transfer cost is low.**, which is formally defined in Section 3.3, where our evaluation metrics is designed with efficiency considerations. $(ii)$ **The task transfer scenarios are broad**. We illustrate and compare CVinW with other settings using a 2D chart in Figure 2, where the space is constructed with two orthogonal dimensions: input image and output concept. The 2D chart is divided into four quadrants, based on how the model evaluation stage is different from model development stage. Both training and evaluation distributions are consistent in both dimensions for the traditional close-set recognition. Open-set recognition allows new concepts in evalua-

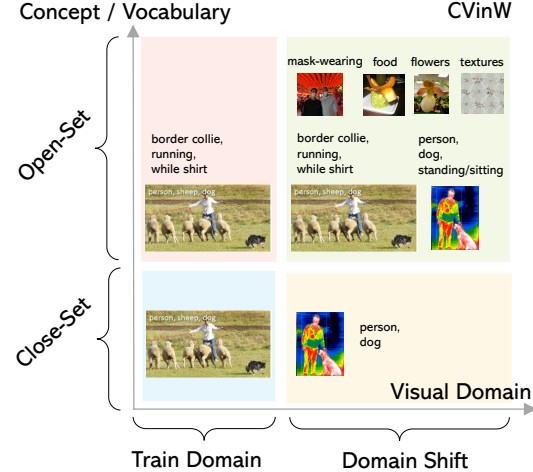

Figure 2: Illustration of CVinW in comparison with close-set, open-set and domain shift.

tion, while typically remains the same visual domain [87, 93]; Domain shift allows new visual domain in evaluation, while typically remains the same concept pool [63, 28]. CVinW allows the flexibility in both dimensions, where any new tasks/datasets in the wild essentially fall into.

# 3 Benchmarks

## 3.1 A Suite of Datasets with Language/Knowledge Augmentations

As a proxy for performing unseen tasks in the wild, we collect a diverse set of public datasets from various domains in computer vision, as the basis of our benchmark. Specifically, we consider 20 datasets for IC and 35 datasets for OD. We exhibit the dataset names in Figure 3 (a), and the detailed statistics of each dataset in Table 5 and Table 6 in Appendix. It is recommended in [66] that studying task-level zero-shot transfer is a way of measuring the task learning capabilities of machine learning systems. The task definition of each downstream recognition dataset is typically specified using category names. Adding user specification/note is a natural way to clarify the task definition *e.g.,* the attribute or explanation of a visual concept. Importantly, a similar spirit has been implemented

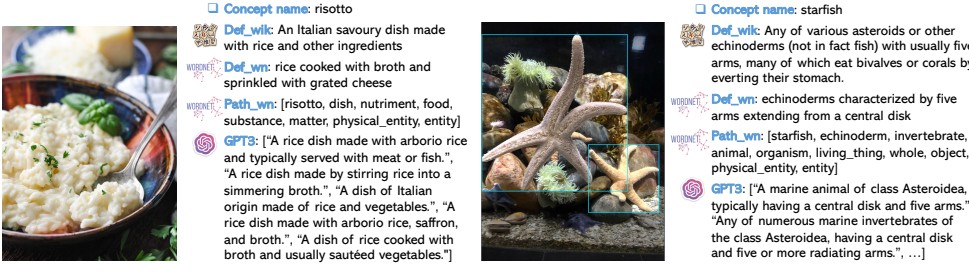

(a) Dataset names. The font size is proportional to the number of concepts in each dataset.

**Concept name:** risotto

**Def_wik:** An Italian savoury dish made with rice and other ingredients

**Def_wn:** rice cooked with broth and sprinkled with grated cheese

**Path_wn:** [risotto, dish, nutriment, food, substance, matter, physical_entity, entity]

**GPT3:** ["A rice dish made with arborio rice and typically served with meat or fish.", "A rice dish made by stirring rice into a simmering broth.", "A dish of Italian origin made of rice and vegetables.", "A rice dish made with arborio rice, saffron, and broth.", "A dish of rice cooked with broth and usually sautéed vegetables."]

**Concept name:** starfish

**Def_wik:** Any of various asteroids or other echinoderms (not in fact fish) with usually five arms, many of which eat bivalves or corals by everting their stomach.

**Def_wn:** echinoderms characterized by five arms extending from a central disk

**Path_wn:** [starfish, echinoderm, invertebrate, animal, organism, living_thing, whole, object, physical_entity, entity]

**GPT3:** ["A marine animal of class Asteroidea, typically having a central disk and five arms.", "Any of numerous marine invertebrates of the class Asteroidea, having a central disk and five or more radiating arms.", …]

(b) Examples of collected external knowledge.

Figure 3: Illustration of our benchmark. Left: Image classification, Right: Object detection.

in traditional class-level zero-shot by adding individual domain-specific knowledge (see Table 1), and demonstrated promising zero-shot performance gains. In this paper, we generalize the notion of "zero-shot" to task-level, collecting external knowledge from general sources for our benchmark.

- *WordNet Hierarchy* (`def_path`). The words along the traversal path from the query node in WordNet [56] to the highest parent node is recorded as the hierarchy knowledge.

- *WordNet Definition* (`def_wn`). The definition in WordNet synsets [56] is used to explain the query.

- *Wiktionary Definition* (`def_wik`). The definition of a query in Wiktionary [55] is used.

- *GPT3 Knowledge* (`gpt3`). For the above three knowledge sources, it is not always feasible to retrieve valid knowledge for any query. To enable full knowledge coverage, we propose to use GPT3 [6] to generate "pseudo" knowledge using in-context-learning, where prompts are constructed with multiple pairs of class names and their Wiktionary definitions. We generate five GPT3 knowledge sequences for each class name, by constructing different context prompts with randomly sampled pairs. See details in Section C.4.

In Fig. 3 (b), we show examples to illustrate the knowledge sources. In practice, there is a trade-off between the knowledge quality and its coverage. For example, WordNet has relatively rich and precise knowledge, but the coverage is low; GPT3 knowledge has the full coverage (as it is generated via prompting a pre-trained neural language model), but it is hard to assess its quality. In the experiment section, we provide baseline results to demonstrate the benefits of external knowledge, and encourage the community to design advanced prompting techniques to leverage these knowledge sources.

### 3.2 Pre-trained Models for Transfer Learning

**Industry Track and Academic Track.** Our benchmark is an evaluation platform for pre-trained models, whose performance largely depends on the scale of the pre-training corpus. Larger corpus typically yields higher performance, but unfortunately results in a barrier to many participants, especially a majority of researchers from university labs. To increase inclusivity, we create two tracks with restrictions on the pre-training data scale: ($i$) *Academic track* is a setting that limits the data in established public large datasets (*i.e.,* ImageNet-21K [14], GCC3M [70] & 12M [8], YFCC15M [78]). This track is more academic-friendly, aiming to encourage the exploration in data-efficient pre-training methodologies. ($ii$) *Industry track* has no limit on pre-training data scale, except that images in our benchmark are not allowed in pre-training when reporting zero-/few-shot performance. This track aims to explore the scaling limit. We encourage participants to report the pre-training datasets to enable reproducible research.

**Pre-trained Models.** To establish baseline results on ELEVATER, we evaluate the pre-trained model checkpoints in Table 2. More details of the checkpoints are described in Appendix. Most existing visual models are language-free, where no text is used in model training. Till recently, visual

| Checkpoints | Taxonomy | | Pre-training Settings | | Network Architecture | | |
|---|---|---|---|---|---|---|---|
| | Language | Knowledge | Training Objective | Dataset | Vision | Language | Others |
| **Image Classification** | | | | | | | |
| MoCo-v3 [10] | ✗ | ✗ | Self-Supervised | ImageNet-1K (1.2M) | ViT-B | - | - |
| MAE [30] | ✗ | ✗ | Self-Supervised | ImageNet-1K (1.2M) | ViT-B | - | - |
| DeiT [79] | ✗ | ✗ | Supervised | ImageNet-1K (1.2M) | ViT-B | - | - |
| ViT [18] | ✗ | ✗ | Supervised | ImageNet-22K (14M) | ViT-B | - | - |
| CLIP [66] | ✓ | ✗ | Image-Text Contrast | WebImageText (400M) | ViT-B | T-B | - |
| UniCL [88] | ✓ | ✗ | Image-Text Contrast | ImageNet-21K (13M) | Swin-T | T-B | - |
| K-LITE [71] | ✓ | ✓ | Image-Text Contrast | ImageNet-21K (13M) | Swin-T | T-B | - |
| **Object Detection** | | | | | | | |
| DyHead [13] | ✗ | ✗ | Supervised | Object365 | Swin-T | - | - |
| GLIP [47] | ✓ | ✗ | Supervised | Object365 & Grounding | Swin-T | Bert-B | Fusion |
| GLIP-A [47] | ✓ | ✗ | Supervised | Object365 | Swin-T | Bert-B | - |
| K-LITE [71] | ✓ | ✓ | Supervised | Object365 | Swin-T | Bert-B | - |

Table 2: The pre-trained models evaluated in ELEVATER as baselines. In terms of taxonomy, ✓ indicates the model checkpoint is pre-trained with the use of language / knowledge, while ✗ indicates language- / knowledge-free. For image classification, the number of images in pre-training is reported. T-B indicate a Base-size Transformer architecture, using a 63M-parameter 12-layer 512-wide model with 8 attention heads. Swin-T is a Tiny-size Swin Transformer [52], Bert-B is a Base-size Bert [16], and Fusion indicates a cross-attention module to fuse the image-text features [47].

models are trained in a language-augmented and/or knowledge-augmented manner using a language model [66, 88, 71, 47], among which CLIP [66] represents a strong baseline in the industry track. Please see the detailed taxonomy in Appendix Section G.1.

### 3.3 Evaluation Settings: Efficiency Considerations

One major advantage of pre-trained models is the promise that they can transfer to downstream tasks *effortlessly*. The cost is considered in two orthogonal dimensions: sample-efficiency and parameter-efficiency, as illustrated in Figure 4. The bottom-left corner and top-right corner is the most inexpensive and expensive adaptation strategy, respectively. One may interpolate and make combinations in the 2D space, to get different model adaptation methods with different cost.

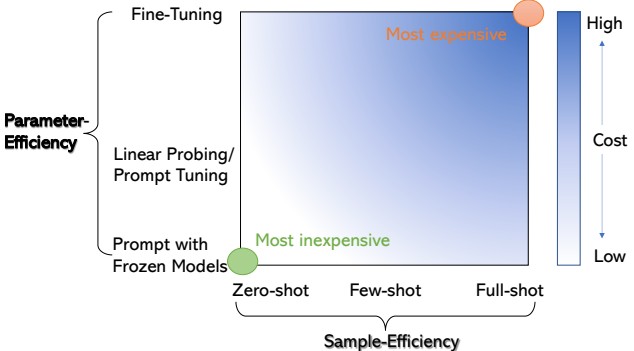

Figure 4: The model adaptation cost chart.

**Sample-efficiency: Zero-, Few-, and Full-shot.** Due to the high cost of annotating data, it is often desired to provide a small number of labeled image-label pairs in downstream datasets. Transferable models should be able to reach high performance in this data-limited scenario. To assess this ability, we vary the number of training set size $N$ per category in the downstream dataset. For IC, $N = 0, 5, 20, 50$. For OD[2], $N = 0, 1, 3, 5, 10$. Three random seeds are chosen, each of which identifies a subset of samples from the full dataset in a deterministic manner. Once the random seed is given, the indices of training samples in few-shot settings are fixed to encourage reproducible research. We also consider the full-shot setting, where all samples of a given dataset are used.

**Parameter-efficiency: Linear Probing vs Full Model Fine-tuning.** Maintaining a small number of dataset-specific model parameters is often favored for model maintenance, as it can be expensive to maintain a unique copy of large model checkpoints for each of the thousands of downstream applications. In IC, linear probing provides a simple strategy for training a dataset-specific linear embedding matrix, while keeping the pre-trained visual backbone frozen. It arguably represents the minimum cost solution for parameter-efficiency. In contrast, fine-tuning often updates the entire weights in backbone and linear head, representing the most expensive solution to model adaptation.

---

[2]For OD, $N$-shot means providing at least $N$ images per category[85, 47]

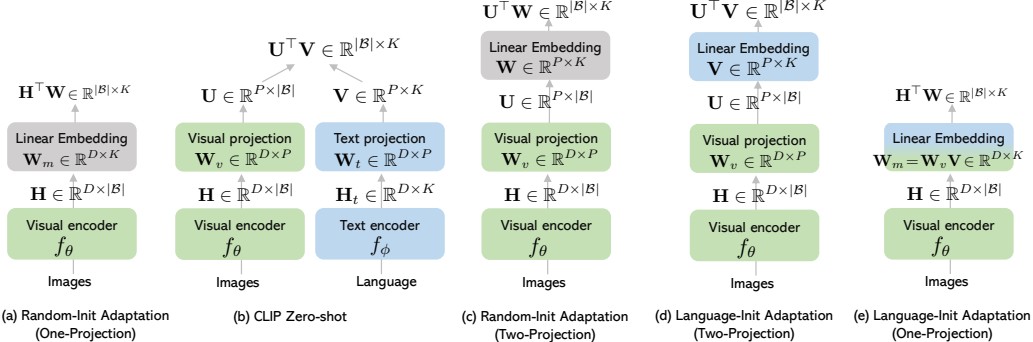

Figure 5: Illustrative comparison of different model evaluation and adaption methods.

In OD [85], linear probing means updating the linear heads for classification and localization tasks only, while fine-tuning means updating all model weights including the backbone and the detectors.

## 4 Toolkits

To ease the process to onboard new checkpoints for evaluation, we provide a software toolkit, including $(i)$ automatic hyper-parameter tuning and $(ii)$ various strategies for model adaptation to downstream tasks. First, automatic hyper-parameter tuning pipeline is developed to avoid human-in-the-loop tuning, thus reducing human labor and ensuring fair comparisons of different model checkpoints. We follow CLIP [66] to implement a simple grid-search style tuning pipeline, and leave more sophisticated methods like BOHB [20] and DEHB [3] as future work. Details are provided in Appendix. Second, we provide several model adaptation methods as strong baselines, which allow effective transfer learning of pre-trained visual models. The ideas are illustrated in Figure 5. For a downstream dataset, we first represent it in a triplet-wise data format $\mathcal{D} = \{(\boldsymbol{x}_n, \boldsymbol{t}_n, y_n)\}_{n=1}^N$, where $\boldsymbol{x} \in \mathcal{X}$ is the image, $\boldsymbol{t} \in \mathcal{T}$ is its corresponding language description, and $y \in \mathcal{Y}$ is a label indicating the index of the unique language description in the dataset. $|\mathcal{B}|$ is batch size. In IC, the number of labels $|\mathcal{Y}| = K$, *i.e.,* the number of category names.

**Language-free Visual Models.** Most existing visual models are language-free, as language is often not considered in training, *e.g.,* supervised and self-supervised methods. Such models can not be directly used for zero-shot transfer, and model adaptation is often enabled by adding additional weights. For each image $\boldsymbol{x}$, an image encoder $f_{\boldsymbol{\theta}}$ parameterized by $\boldsymbol{\theta}$ first represents $\boldsymbol{x}$ as a visual feature vector $\boldsymbol{h} \in \mathbb{R}^{D \times 1}$: $\boldsymbol{h} = f_{\boldsymbol{\theta}}(\boldsymbol{x})$. One randomly initialized linear projection layer with $\mathbf{W}_m \in \mathbb{R}^{D \times K}$ (we absorb the bias $\boldsymbol{b}$ in $\mathbf{W}_m$ for simplicity) is used as the classifier; see Figure 5 (a).

**Language-augmented Visual Models.** Recent works [66, 35] that learn visual models with language supervision often employ a two-encoder architecture. Besides the image encoder model $f_{\boldsymbol{\theta}}$, a text encoder $f_{\boldsymbol{\phi}}(\boldsymbol{t})$ parameterized by $\boldsymbol{\phi}$ is also used to encode text $\boldsymbol{t}$. Additional projection layers $\mathbf{W}_v$ and $\mathbf{W}_t$ are introduced for image and language features, embedding them into a joint space with dimension $P$, with projected features as $\boldsymbol{u}$ and $\boldsymbol{v}$ respectively. Note that lowercase $\boldsymbol{u}$ and $\boldsymbol{v}$ are single feature vectors while $\mathbf{U}$ and $\mathbf{V}$ are a batch with multiple feature vectors. As in Figure 5 (b), zero-shot learning can be directly performed in this space: the mean text feature $\boldsymbol{v}$ is first obtained for each category, by averaging text features of the category name in different language prompts. The image is predicted as the category yielding the highest similarity $\boldsymbol{u}^\top \boldsymbol{v}$.

- *Random initialized Adaptation.* In the original CLIP paper [66], one randomly initialized linear projection layer $\mathbf{W} \in \mathbb{R}^{P \times K}$ (similarly, we absorb the bias $\boldsymbol{b}$ in $\mathbf{W}$ for simplicity) is added on the pre-retained visual projection, which is shown as the two-projection method in Figure 5 (c).

- *Language-initialized Adaptation.* We argue that the full capacity of language-augmented visual models is not leveraged in [66]. The power of pre-trained language encoder and text inputs must play a vital role in model adaptation. Hence, we propose two language-initialized adaption methods, each of which is ensured as a fair comparison variant for language-augmented and language-free models, respectively. $(i)$ *Two-Projection.* For the linear head $\mathbf{W} \in \mathbb{R}^{P \times K}$ added on the projection space, we initialize $\mathbf{W}$ with $\mathbf{V}$ (bias terms are initialized as zeros), as shown in Figure 5 (d). In this way, the visual and text heads are separated. Note that the language-initialized Two-Projection

scheme is also basically equivalent to Figure 5 (b) in zero-shot settings. Please see Appendix Section F for discussions. $(ii)$ *One-Projection*. To fairly compare with language-free model adaptation in Figure 5 (a), one linear projection head should directly be added on the backbone (before the visual projection) to ensure that the same number of trainable parameters are updated. Therefore, we propose to initialize $\mathbf{W}_m \in \mathbb{R}^{D \times K}$ in this case with the multiplication result of two linear matrices $\mathbf{W}_v \mathbf{V}$, as shown in Figure 5 (e).

**Discussion.** We highly recommend the proposed language-initialized methods as the standard to adapt language-augmented visual models for two reasons: $(i)$ This simple method yields surprisingly superior empirical performance, as demonstrated in our experiments. $(ii)$ It provides an effective mechanism to leverage the external knowledge that is collected for a downstream task in our benchmark. Specifically, the knowledge can be concatenated with the original language prompt (with a simple ";" in our experiments), then encoded into contextualized text features. When multiple knowledge items exist (*e.g.,* the case of GPT-3) for each concept, we concatenate one of its prompts and one of its knowledge items, and get the encoded text embedding of the concatenated sequence via the language encoder. This is performed for all the combinations between all prompts and knowledge items for this concept, then the averaged embedding is computed to represent the concept. In contrast, random initialization would ignore this knowledge source. The language-initialized method can serve as a strong baseline to encourage more effective knowledge-augmented adaptation methods.

In OD, GLIP is a language-augmented detector, whose overall architecture can be simply considered as adding a cross-modal module over the CLIP-like dual-encoder. In GLIP [47], its linear probing has been implemented via updating $\mathbf{W}_v$ and $\mathbf{W}_t$. A prompt-tuning strategy was proposed, by initializing the language input of the cross-modal module as $\mathbf{V}$, and only updating $\mathbf{V}$ during adaptation. This is similar to our language-initialized strategy.

## 5 Empirical Results and Findings

We present the experimental results with our benchmark to illustrate two points. Q1: The importance of language in visual model transfer in the adaptation stage. Q2: We present three playgrounds that our benchmark can help to cultivate research in, including sample-efficiency, parameter-efficiency and external knowledge for visual transfer. We also present novel empirical findings.

### 5.1 The Role of Language for Vision

**Effectiveness of Language-initialized Adaptation Methods.** In Table 3, we compare the effectiveness of the proposed language-initialization methods with the checkpoint CLIP ViT-B32. The one-projection scheme is consistently better than two-projection scheme in all settings (though the gain is minor). This is because the former often has less parameters than the latter, as $D = 768 > P = 512$. To ensure fair comparisons with the random initialization of linear head in CLIP [66] (*i.e.,* # trainable parameters is the same), in the ensuing experiments, we consider the two-projection language-initialization scheme as the default, unless the one-projection scheme is specified.

As shown in Fig. 6, under both linear probe (LP) and fine-tuning (FT) settings, language-based initialization significantly outperforms random initialization. Notably, we show that even with very few shots (*e.g.,* 2-shots), both our LP and FT is able to outperform the zero-shot CLIP. This is contradictory to the finding in the original CLIP paper [66], where zero-shot outperforms linear probing in the fewer shot (less than 4) settings. With the proposed language-init method, one can ensure that few-shot performance is always better than zero-shot, as we essentially reduce to zero-shot when zero iteration is updated in our language-init method. Moreover, we also find that with random initialization, FT performs significantly worse than LP under few-shot settings. However, with language-init, FT starts to outperform LP with more than 20 shots. Both findings demonstrate the proposed language-based initialization is consistently effective, suggesting that it is an important

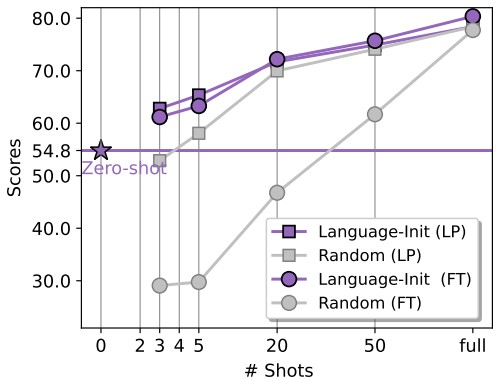

Figure 6: Comparison of random- and language-initialized adaptation.

| Checkpoint | Settings | | 20 IC datasets | | |
|---|---|---|---|---|---|
| | Adaptation | Initialization | Zero-shot[†] | Few-shot (5, 20, 50) | Full |
| **Industry Track** *(No pre-train data scale limit)* | | | | | |
| CLIP (ViT-B32) | LP | Random-2P | 56.64 | $58.09_{\pm 2.80}$, $69.97_{\pm 1.30}$, $74.09_{\pm 0.69}$ | 78.38 |
| | LP | Language-2P | | $65.35_{\pm 1.24}$, $71.69_{\pm 0.93}$, $74.89_{\pm 0.79}$ | 78.40 |
| | LP | Language-1P | | $65.88_{\pm 0.79}$, $72.05_{\pm 0.85}$, $75.08_{\pm 0.73}$ | 78.96 |
| | FT | Random-2P | | $29.75_{\pm 6.64}$, $46.76_{\pm 11.9}$, $61.70_{\pm 9.97}$ | 77.77 |
| | FT | Language-2P | | $63.29_{\pm 3.18}$, $72.19_{\pm 1.31}$, $75.70_{\pm 1.14}$ | 80.35 |
| Supervised (ViT-B32) | LP | Random-1P | - | $56.00_{\pm 2.67}$, $67.23_{\pm 1.66}$, $71.35_{\pm 1.17}$ | 75.29 |
| | FT | Random-1P | | $58.55_{\pm 2.58}$, $71.27_{\pm 1.25}$, $75.36_{\pm 1.42}$ | 80.39 |
| **Academic Track** *(Pre-trained on large established public datasets)* | | | | | |
| UniCL (Swin-Tiny) | LP | Language-2P | 27.15 | $54.31_{\pm 4.15}$, $66.42_{\pm 2.08}$, $70.49_{\pm 1.01}$ | 74.75 |
| | FT | Language-2P | | $44.75_{\pm 5.42}$, $56.53_{\pm 5.37}$, $67.90_{\pm 5.31}$ | 78.48 |
| K-LITE (Swin-Tiny) | LP | Language-2P | 33.44 | $55.06_{\pm 2.36}$, $66.26_{\pm 1.56}$, $70.16_{\pm 1.09}$ | 74.47 |
| | FT | Language-2P | | $48.41_{\pm 2.84}$, $58.06_{\pm 4.30}$, $71.66_{\pm 2.02}$ | 78.05 |

Table 3: Averaged results on 20 IC datasets using linear probing (LP) and fine-tuning (FT). Random-1P, Random-2P, Language-1P and Language-2P indicates the initialization method in Figure 5 (a), (c), (e) and (d), respectively. † Note that one zero-shot result is reported for each model checkpoint using the method in Figure 5 (b), which is independent from adaptation/initialization methods.

| Checkpoint | Adaptation | 35 OD datasets | | |
|---|---|---|---|---|
| | | Zero-shot | Few-shot (1, 3, 5, 10) | Full |
| GLIP (Swin-Tiny) | Prompt | 19.7 | $29.7_{\pm 0.4}$, $36.5_{\pm 0.6}$, $39.0_{\pm 1.1}$, $41.8_{\pm 1.2}$ | 54.4 |
| | LP | | $22.2_{\pm 0.1}$, $24.4_{\pm 0.2}$, $25.1_{\pm 0.2}$, $25.6_{\pm 0.6}$ | 35.2 |
| | FT | | $32.2_{\pm 0.7}$, $39.2_{\pm 0.3}$, $42.5_{\pm 0.9}$, $49.1_{\pm 0.6}$ | 63.2 |
| DyHead (Swin-Tiny) | LP | - | $15.2_{\pm 0.6}$, $19.2_{\pm 0.9}$, $19.8_{\pm 1.0}$, $20.6_{\pm 1.1}$ | 31.4 |
| | FT | | $25.6_{\pm 0.4}$, $37.1_{\pm 0.5}$, $40.1_{\pm 1.5}$, $44.6_{\pm 0.7}$ | 63.9 |

Table 4: Averaged results on 35 OD datasets.

technique, and should be the standard adaptation method for language-augmented visual models like CLIP. Further, the correct adaptation methods for language-augmented visual models should leverage both the pre-trained visual and text encoder. It is not sufficient to solely transfer from the visual encoder, pre-trained language encoder plays an important role in task transfer.

**The Competition of Pre-trained Models: Language-free vs Language-augmented.** We summarize the transfer performance of pre-trained models for IC in Table 3 and OD in Table 4. For IC, we also compare CLIP against other language-free visual models including MoCov3, MAE, ViT, DeiT in Appendix. We see that the language-augmented model (CLIP) outperforms language-free model (Supervised ViT) in the limited data settings. The gap is closed when more training examples are observed (*e.g.,* , 50-shot and full-shot). This is probably because the pre-training power is gradually dominated by larger-scale downstream training. Further, language-augmented models are able to perform zero-shot task transfer, while traditional language-free models cannot. Similar conclusions can be drawn for OD in Table 4. Hence, we recommend the use of language-augmented visual models for task-level transfer.

### 5.2 Playground I: Sample Efficiency
We explore sample efficiency in Fig. 7 (a) for IC. First, we find that CLIP consistently outperforms supervised ViT (Sup-ViT), yielding a significant 5~10% gain in the 5-shot settings. This suggests that CLIP is more sample-efficient than supervised ViT . Furthermore, we find that fine-tuning CLIP yields better performance than linear probing in >20-shot settings, while being worse in the 5-shot setting. This is a bit surprising, as it is contradictory to the common convention that fine-tuning is always better than linear probing. We hypothesize this is because fine-tuning tends to over-fit in the scenarios with a large number of trainable parameters and a small number of training samples. Overall, it suggests that fine-tuning CLIP potentially has a better sample efficiency than linear probing, and a better adaptation strategy on fewer-shot settings can be explored in the future. For supervised ViT, FT is always better than LP, the performance gap becomes larger when more samples are used. In 5-shot settings, the gap is minor, which is similar to observations made for supervised CNNs [91, 37, 38]. To compare pre-trained models, we suggest to report the evaluation results on the entire spectrum of sample-efficiency to fully study the behaviors of a pre-trained model. If compute resource is limited, zero-shot or few-shot evaluation can be used as a quick assessment.

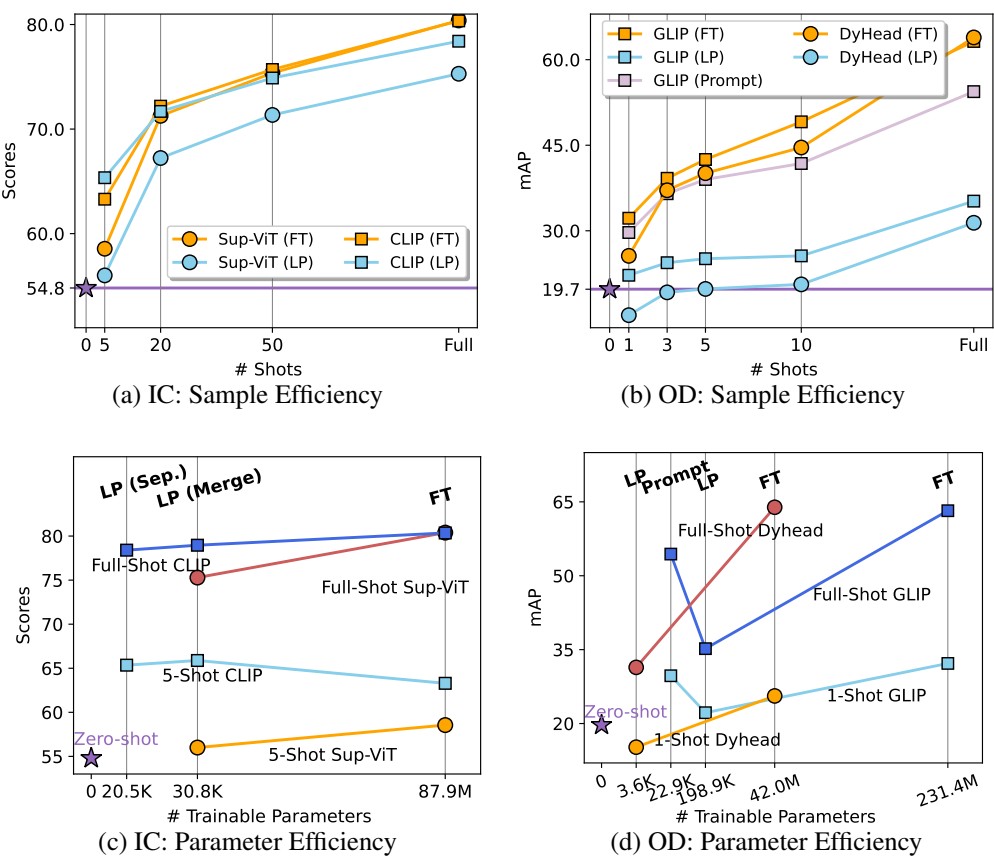

| (a) IC: Sample Efficiency | (b) OD: Sample Efficiency |
|---|---|

| (c) IC: Parameter Efficiency | (d) OD: Parameter Efficiency |
|---|---|

Figure 7: Adaptation efficiency considerations. For IC, comparison of adaptation efficiency between CLIP and supervised ViT (Sup-ViT). For OD, comparison of adaptation efficiency between GLIP and DyHead.

We explore sample efficiency for OD in Fig. 7 (b). The conclusion is similar to IC in that the language-augmented visual model (GLIP) is more sample-efficient than the language-free visual model (DyHead), when the models are adapted using either LP or FT settings. The performance gap is large in the fewer-shot settings and is small in the full-shot settings. The difference is that FT consistently outperforms LP in all settings for OD. This is probably because there are a lot of boxes (training instances) per image in OD, which makes OD less likely to over-fit compared to IC.

## 5.3 Playground II: Parameter Efficiency

We study the parameter efficiency in Fig. 7 (c) for IC. For CLIP, we experiment with two different settings of linear probing on whether to merge the last two linear projection layers ($\mathbf{W}_v$ and $\mathbf{V}$ in Fig. 5). Merging these two layers in CLIP allows $1.5\times$ trainable parameters in the linear probe classifier as keeping them separated. First, it shows a trend that a larger number of trainable parameters leads to better performance, as demonstrated by three curves/scenarios: full-shot CLIP, full-shot Sup-ViT and 5-shot Sup-ViT. This also verifies that LP and FT provide the lower bound and upper bound, respectively, in terms of both #parameter and performance. Most existing parameter-efficient adaptation methods play a trade-off game. However, in the scenario of 5-shot CLIP, we do notice a slight drop in performance when we further increase the number of trainable parameters to full-model fine-tuning. It suggests that the scenario of adapting language-augmented visual models for data-limited settings is a more meaningful playground to explore the line of research in parameter-efficient adaptation methods, as the best performance may require an optimal number of trainable parameters, which has been less explored.

We study the parameter efficiency in Fig. 7 (d) for OD. The overall trend is similar in that better performance comes with more parameters. It turns out that prompt tuning an language-augmented OD model is an effective parameter-efficient approach. For example, prompting is better than linear

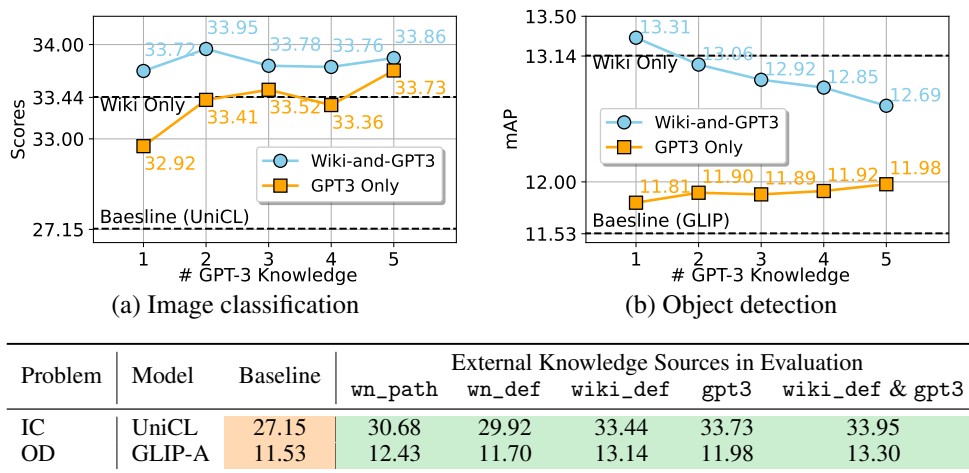

|           |     |            | (a) Image classification | (b) Object detection |

| Problem | Model | Baseline | External Knowledge Sources in Evaluation | | | | |
|---------|-------|----------|----------|---------|----------|------|----------------|
|         |       |          | wn_path  | wn_def  | wiki_def | gpt3 | wiki_def & gpt3 |
| IC      | UniCL | 27.15    | 30.68    | 29.92   | 33.44    | 33.73 | 33.95         |
| OD      | GLIP-A | 11.53   | 12.43    | 11.70   | 13.14    | 11.98 | 13.30         |

Figure 8: Zero-shot task transfer with various external knowledge sources in the evaluation stage. In (a) and (b), a varying number of generated GPT-3 knowledge sequences is utilized for inference, and "Wiki-and-GPT3" indicates both Wiktionary and GPT3 knowledge are used simultaneously. The bottom table summarizes the prediction result for each knowledge source.

probing in GLIP. Further, prompt tuning GLIP outperforms fine-tuning DyHead in the 1-shot setting, where the former has less than 0.1% parameters of the latter.

### 5.4 Playground III: The Benefit of External Knowledge for Vision

We investigate the effectiveness of external knowledge in Fig. 8, measured by zero-shot task transfer performance. The model K-LITE is evaluated, as its pre-training is knowledge-augmented. We find that leveraging external knowledge improves upon the knowledge-free pre-training counterparts (UniCL and GLIP). For example, UniCL is improved from 27.15 to 29.92~33.93, and GLIP-A is improved from 11.53 to 11.70~13.30. Further, for GPT3 knowledge, a larger number of generated knowledge items often leads to higher performance. When combining GPT3 knowledge with Wiktionary knowledge, we see a further performance boost. With an increasing number of GPT3 knowledge items, the gain is consistently improved for IC, but not for OD. In Table 3, we study the role of knowledge for task transfer in model adaptation. Initializing the linear head using features encoded with knowledge is an effective way to leverage the collected knowledge sources, especially for the fewer-shot settings.

One may wonder if the collected knowledge in ELEVATER benchmark can also benefit knowledge-free pre-trained models such as CLIP during model adaptation? We confirm its effectiveness in Appendix. In zero-shot transfer, external knowledge improves the baseline on four datasets. In few- and full-shot transfer, one may selectively choose whether to update the model using external knowledge, by observing the best performance in the auto-tuning stage. This selective strategy with the availability of external knowledge demonstrates a consistent improvement (or tie) on 15+ out of 20 datasets.

## 6 Conclusions

We have presented ELEVATER, a platform to evaluate the recently emerging language-augmented visual models for task-level transfer. It consists of 20 image classification datasets and 35 object detection datasets. All of them are collected from public domains, and are enriched with various external knowledge sources to enhance the language modality. We have developed open-source toolkits with an auto hyper-parameter pipeline and novel language-initialized adaptation methods to ensure easy utilization and fair comparisons. Strong baseline results are produced from the toolkit to cultivate research in a variety of topics, *e.g.,* more transferable language-augmented visual models, advanced model adaption methods (sample-efficiency and parameter-efficiency), and external knowledge for task-level transfer. The question of how to design general-purpose task-level transferable visual models remains largely unanswered. Given benchmarks and tookits we have developed from the perspective of language-augmented visual models, we believe that ELEVATER can provide fertile soil for addressing this challenge.

## Acknowledgments

The authors gratefully acknowledge Haotian Zhang for building the ODinW leaderboard on Eval AI, Pengcheng He for helpful discussions to have a separate track dedicated for users from academia, Baolin Peng and and Zhengyuan Yang for the inspirations of GPT3 to generate knowledge for dialogue and OK-VQA tasks, Bo Li for insights on the topic of domain generalization, Zhuowen Tu for the inspirations to make benchmark scope wider to measure all pre-trained vision models, Ce Liu for suggestions to compare the benchmark with well-established vision datasets such as ImageNet and COCO. The benchmark depends on publicly available datasets; we acknowledge all the original authors who made their datasets public. Please follow the original license of each dataset and keep this benchmark for academic purposes. This work was supported in part by NSF CAREER IIS-2150012, the Wisconsin Alumni Research Foundation, and Institute of Information & communications Technology Planning & Evaluation(IITP) grants funded by the Korea government(MSIT) (No. 2022-0-00871, Development of AI Autonomy and Knowledge Enhancement for AI Agent Collaboration) and (No. RS-2022-00187238, Development of Large Korean Language Model Technology for Efficient Pre-training).

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
