# OpenReview forum: "ELEVATER: A Benchmark and Toolkit for Evaluating Language-Augmented Visual Models"
_NeurIPS.cc/2022/Track/Datasets_and_Benchmarks — NeurIPS 2022 Datasets and Benchmarks _

### Official Review · Reviewer_onGS · 2022-07-07
**Review of ELEVATER : A Benchmark and Toolkit for Evaluating Language-Augmented Visual Models**

**Rating:** 7
**Confidence:** 4

**Strengths:**

The authors propose a way to use language information called "language-init" for downstream learning. This approach showed that using language+visual information downstream improves performance on few-shot and zero-shot learning. This is an interesting approach that does not rely on prompt engineering.

The authors run comprehensive evaluation on a variety of language-based models and some visual-only models. They also explore pre-training strategies on the language side. They provide some interesting insights on how language can be used in downstream learning for visual-only tasks.


**Weaknesses:**

While Figure 2 is sufficient, it would be useful to have an overall visual depiction of the evaluation pipeline. While the paper is well written, it is still a little difficult to follow and a visualization of the overall process may help.

It is not clear to me how the external knowledge is used for evaluation. I understand you are using it to generate descriptions of a visual object for example, but it is not clear how that is used in the evaluation. For example, in Section 5.4 "We find that leveraging external knowledge improves upon the knowledge-free pre-training counterparts (UniCL and GLIP)."  in lines 286-287 but there is no clear explanation of how you are leveraging external knowledge. Is this just during pre-training? And if it is used in pre-training, it is not clear how.

The paper leads you to believe that there would also be an including of text analysis in this benchmark. You ask how using language improves visual learning, but why not the reverse?

There is a lack of introducing prompting. Even though I am very familiar with prompting for zero-shot learning, I did not truly understand some of the details of the prompting strategy used until I explored the repositories related to this paper.

Overall, my major concern is the lack of clarity on the approaches. The prompting strategy used is extremely unclear in both the paper and the appendix, there is too strong an assumption that the reader should have full understanding of CLIP and prompting. The pre-training strategies are also confusing although more clear in the appendix.

**Additional Feedback:**

Table 1 should be more clear on what IC and OD are in case someone is skimming for information, especially because the table provides a great deal of information readers will refer to.

Figure 3 should be more specific to what LP and FT are for skimming.

In section 5, you propose two "research questions", but they are not actual research questions, more like statements of what you are addressing.

As mentioned earlier, my main complaint is the clarity of the benchmark in relation to prompting, pre-training and the use of external knowledge.


**Clarity:**

The paper is very well written but the content can be quite confusing. As mentioned earlier, some of the confusing items are:
1) How external knowledge is used was not clear in neither the paper. Is it for prompting? Is it for pre-training? I know it is in more detail in the appendix, but it should be clearer in the main paper as well.
2) That the focus was on downstream visual tasks not joint visual-text evaluation.
3) Whether they were using existing pre-trained models, pre-training themselves via the toolkit or some combination of.
4) How is prompting used, is external knowledge included in the engineering or are you using existing methods for zero-shot prompting?


**Correctness:**

This is the first toolkit to my knowledge. Others are related to visual reasoning tasks for visual-text models.

The evaluation techniques and experimental designs were performed correctly and explained in great detail.

**Documentation:**

The documentation is good in both the paper, the website and the repositories.

However, the object detection toolkit appears incomplete and is not a toolkit in its current state.

**Ethics:**

I do not know if this is the appropriate section, but it was very considerate to make both academic and industry tracks in order to increase  inclusivity to all levels of accessibility.

**Relation To Prior Work:**

There was no reference to any previous benchmarks that include the text domain of visual question answering, video captioning, etc. I think it would be beneficial to discuss these in isolation and how this work is different or addresses limitations to existing ones.

I also think prompting should be mentioned for zero-shot learning in the related works.

**Summary And Contributions:**

ELEVATER is a benchmark toolkit for evaluating visual and text models. For image classification, it is an aggregation of 20 datasets with 1151 concepts and 3 text external knowledge bases. It allows for full, zero-shot, and few-shot evaluation. For object detection, it is an aggregation of 35 datasets and 314 concepts with 3 text external knowledge bases. Authors claim to be the first public
benchmark to standardize the zero-shot task-level transfer setting. They also provide a hyperparameter tuning toolkit to help evaluation. Using the benchmark, they have shown that it is not enough to just transfer from the visual encoder but the language-encoder should also be utilized. In summary, the authors provide a toolkit that automates hyperparameter tuning for downstream image classification and object detection. They evaluate 1) how language-visual models perform in comparison to visual-only models 2) how (I think) pre-training with additional language information improves downstream performance 3) and how different downstream approaches like FT, LP and using language in addition to visual change performance. I would consider their major contributions are their "language-init" approach and their comprehensive model comparisons as shown in the Supplementary for paper content and the toolkit for overall.

---

> ### Author Response · Authors · 2022-08-23
> **Response to Reviewer onGS (Part 1)**
>
> ## Q & A on Section ``Weaknesses'':
>
> > **W1**: While Figure 2 is sufficient, it would be useful to have an overall visual depiction of the evaluation pipeline. While the paper is well written, it is still a little difficult to follow and a visualization of the overall process may help.
>
> A:  The overall visual depiction of the evaluation pipeline is added in Figure 1.
>
> ---
>
> > **W2**: It is not clear to me how the external knowledge is used for evaluation. I understand you are using it to generate descriptions of a visual object for example, but it is not clear how that is used in the evaluation. For example, in Section 5.4 "We find that leveraging external knowledge improves upon the knowledge-free pre-training counterparts (UniCL and GLIP)." in lines 286-287 but there is no clear explanation of how you are leveraging external knowledge. Is this just during pre-training? And if it is used in pre-training, it is not clear how.
>
> A: As a benchmark paper, we only focus on leveraging external knowledge **in the evaluation stage**. Specifically, this is done by:
> - Constructing prompts with external knowledge; the knowledge-augmented prompts are used to represent the concept, instead of knowledge-free prompts.
> - For zero-shot learning, the text embeddings of knowledge-augmented prompts are used for image-to-text similarity comparison;  For few-shot and full-shot, the text embeddings of knowledge-augmented prompts are used for language-initialization model adaptation.
>
> In the literature, to the best of our knowledge, the only knowledge-augmented pre-trained language-image model is the KLITE, all others are knowledge-free pre-trained models, including UniCL/CLIP, GLIP.
>
>
> > **W3**: The paper leads you to believe that there would also be an including of text analysis in this benchmark. You ask how using language improves visual learning, but why not the reverse?
>
> A: Sorry for the confusion. With a focus scope, we only study the scenarios of improving visual learning with language supervision. This itself has lead to a lengthy paper draft (10 pages in main paper and 12 pages in Appendix). We agree that the reverse problem (improving language learning with visual supervision) is interesting and less explored. We discuss it in Related Work (Line 91-95), and leave it as future works.
>
> > **W4**: There is a lack of introducing prompting. Even though I am very familiar with prompting for zero-shot learning, I did not truly understand some of the details of the prompting strategy used until I explored the repositories related to this paper. Overall, my major concern is the lack of clarity on the approaches. The prompting strategy used is extremely unclear in both the paper and the appendix, there is too strong an assumption that the reader should have full understanding of CLIP and prompting. The pre-training strategies are also confusing although more clear in the appendix.
>
> A: We add the details of how to construct prompts with and without external knowledge as follows:
>
> - All the prompt templates are data-specific. They are summarized in one file in our toolkits [vision_benchmark/datasets/prompts.py]. Due to limited space, we decide to provide the link in our paper, instead of copying them into the Appendix.
>
> - To address the details of prompting with and without knowledge, we add Section C4 in Appendix named “Prompting and Knowledge". It provide the step-to-step guidance to illustrate prompting procedure.
>
> - We provide one example in Table 7, to illustrate prompt templates, knowledge, and prompt construction with and without external knowledge for the concept pink primrose in dataset Oxford Flowers 102.
>
> - The description to construct prompts with multiple knowledge items are added in Line 203-210 in the main paper.
>
> Note that this benchmark paper only investigates evaluation strategies with external knowledge, rather than pre-training strategies with external knowledge. Please check out KLITE [*] for pre-training strategies with external knowledge.
>
> [*] K-LITE: Learning Transferable Visual Models with External Knowledge

---

> ### Author Response · Authors · 2022-08-23
> **Response to Reviewer onGS (Part 2)**
>
> *Note: We found that some of the comments from Reviewer onGS are similar across different sections, and thus we decide to provide full response for comments that appear early, and provide brief response with references for comments that appear late.*
>
> ## Q & A on Section ``Clarity'':
>
> > How external knowledge is used was not clear in neither the paper. Is it for prompting? Is it for pre-training? I know it is in more detail in the appendix, but it should be clearer in the main paper as well.
>
> A: Please see our response **W2** and **W4**  in Response to Reviewer onGS (Part 1)
>
> > That the focus was on downstream visual tasks not joint visual-text evaluation.
>
> A: The focus of this benchmark is "improving visual learning with language supervision", but not "joint visual-text evaluation" or "improving language learning with visual supervision". We add a new paragraph in Related Work (Line 91-95) to discuss it.
>
> > Whether they were using existing pre-trained models, pre-training themselves via the toolkit or some combination of.
>
> A: All the checkpoints are from existing pre-trained models. We did not pre-train models.
>
> > How is prompting used, is external knowledge included in the engineering or are you using existing methods for zero-shot prompting?
>
> A: (1) For prompting without knowledge, we followed CLIP paper; (2) For prompting with knowledge, we propose to simply concatenate the original prompt with retrieved knowledge sequence, and thus external knowledge is included in the prompt engineering. Please our response to **W4** in Response to Reviewer onGS (Part 1)
>
> ---
>
> ## Q & A on Section ``Relation To Prior Work'':
>
> > There was no reference to any previous benchmarks that include the text domain of visual question answering, video captioning, etc. I think it would be beneficial to discuss these in isolation and how this work is different or addresses limitations to existing ones.
>
> A: We add a new paragraph in Related Work (Line 91-95) to discuss it.
>
> > I also think prompting should be mentioned for zero-shot learning in the related works.
>
> A: We add some discussion in Line 622-625:``In particular, natural language prompting is the method of reformatting NLP tasks in the format of a natural language response to natural language input, has attracted attentions in zero-shot and few-shot learning in NLP. It has inspired a few recent works for language-augmented visual models''.
>
> We would be happy to discuss any specific papers you have in mind on ``prompting for zero-shot learning''.
>
>
> ---
>
> ## Q & A on Section ``Documentation'':
>
> > However, the object detection toolkit appears incomplete and is not a toolkit in its current state.
>
> A: The link we provided was embedded in the GLIP repo, which is a detection codebase based on maskrcnn. Since the model adaptation stage and model pre-training stage are typically heavily entangled for object detection, we decide to use GLIP as a tutorial to onboard new OD models.  This GLIP repo provides the *full toolkit* to adapt and evaluate OD models on ODinW. We have updated the link [[ODinW Toolkit]](https://github.com/microsoft/GLIP#the-object-detection-in-the-wild-benchmark), and our documentation for completeness.
>
> ---
>
> ## Q & A on Section ``Additional Feedback'':
>
> > Table 1 should be more clear on what IC and OD are in case someone is skimming for information, especially because the table provides a great deal of information readers will refer to.
>
> A: Full names are now provided.
>
> > Figure 3 should be more specific to what LP and FT are for skimming.
>
> A: Full names are in the caption.
>
> > In section 5, you propose two "research questions", but they are not actual research questions, more like statements of what you are addressing.
>
> A: It is revised as "points".
>
> > As mentioned earlier, my main complaint is the clarity of the benchmark in relation to prompting, pre-training and the use of external knowledge.
>
> A: Please see our response **W2** and **W4**  in Response to Reviewer onGS (Part 1). This benchmark paper does not investigates pre-training with external knowledge.

---

### Official Review · Reviewer_Jeow · 2022-07-24

**Rating:** 7
**Confidence:** 3
**Clarity:** The paper is clearly written, and eas…

**Strengths:**

1. The benchmark is well motivated. It is clearly described how it is different from previous work.
2. The  novel industry and academic tracks in the benchmark are a really nice addition, as they make for a more equitable evaluation of vision models.
3. The proposed methods for task adaptation via language initialization are well motivated and show convincing results.
4. The experiments are very thoroughly designed, and the results are convincing. The results are mostly consistent with the hypotheses.
5. The experiments for integrating knowledge into task transfer are well designed, and the use of knowledge is also well motivated.

**Weaknesses:**

1. The choice of aggregating scores across tasks, while standard, is questionable, since equal improvements on different tasks are not equivalent. Aggregating scores does not give us insights into what tasks the improvements are actually seen on. For instance, a 1% improvement on a task with 80% accuracy is not the same as 1% improvement on a 50% accuracy task. Further, tasks with different number of output labels have different baselines, and aggregating the scores does not account for the variance in those tasks' baseline scores.

**Additional Feedback:**

-

**Correctness:**

The aggregation of scores is arguably not wrong, but does not accurately reflect model improvements in my opinion.

**Documentation:**

Yes

**Ethics:**

No issues.

**Relation To Prior Work:**

The paper clearly compares its benchmark against several zero-shot and few-shot image classification datasets/benchmarks, but only compares against a single object detection dataset. More discussion of object detection datasets and benchmarks would be useful.

**Summary And Contributions:**

This paper introduces ELEVATER, a benchmark to evaluate the transferability of vision models to new datasets and tasks. The benchmark collates a large range of image classification and object detection datasets, that pre-trained vision models (such as CLIP and ViT) can be transferred to. The benchmark also contains evaluation protocols for measuring transferability in different n-shot settings, and also suggests language and knowledge augmentation techniques to add the cross-task transfer. The authors perform extensive experiments, using both language-free and language-augmented vision models, to evaluate how these models transfer to new vision tasks in terms of both sample efficiency and tuneable parameter efficiency.

---

> ### Author Response · Authors · 2022-08-23
> **Response to Reviewer Jeow**
>
> > The choice of aggregating scores across tasks, while standard, is questionable, since equal improvements on different tasks are not equivalent. Aggregating scores does not give us insights into what tasks the improvements are actually seen on. For instance, a 1% improvement on a task with 80% accuracy is not the same as 1% improvement on a 50% accuracy task. Further, tasks with different number of output labels have different baselines, and aggregating the scores does not account for the variance in those tasks' baseline scores.
>
> A: This is a very insightful observation, and we agree that simple average is not perfect. As the reviewer rightly pointed out, simple average is ``standard'', for example, the established benchmarks such as GLUE and SuperGLUE in NLP employ this strategy. While we running experiments to study the final metrics, we provide our thoughts as follows for discussions:
>
> Internally, we are discussing various ways for weighted average as the choice of aggregating scores across tasks.
>
> *(a) Model-agnostic*. The weighting factor could be related to:
>  - The number of concepts per dataset: $K$
>  - The number of images in the test set: $|D_{test}|$
>
> Since random guess yields the prediction accuracy: $1/K$, it means larger $K$ typically provides a more difficult task. On the other hand, $|D_{test}|$  are the number of samples for a prediction task, it means larger $|D_{test}|$ typically provides more robust estimate of the task difficulty. A simple strategy is to consider the entropy $ \log K$ (or more precisely, the entropy of the empirical distribution) as the measure of difficulty, and compute the weighted average.
>
> *(b) Model-dependent*. A better strategy in theory is to consider the entropy of prediction distribution $p(y|x)$ with input $x$ and output $y$. However, a computational model is often involved in the estimate of the information measure for high-dimension data, which make it model-dependent, and related evaluation results will arguably be biased to underlying model family.
>
> We tend to design an model-agnostic score aggregating method.
>
> ---
>
> > The paper clearly compares its benchmark against several zero-shot and few-shot image classification datasets/benchmarks, but only compares against a single object detection dataset. More discussion of object detection datasets and benchmarks would be useful.
>
> A: Thanks for the suggestion. We add the discussion on more object detection benchmarks, including COCO, LVIS and UODB. See Table 1 and related discussions.

---

> > ### Comment · Reviewer_Jeow · 2022-08-23
> > **Reviewer Response**
> >
> > Thank you for the response. I think looking at relative improvements on each task vs a random baseline, or making all evaluations a comparison between two models and looking at the relative improvement on each task from one model to another, would be a good solution. In either case, I am satisfied with this paper, and keep the rating at 7.

---

### Official Review · Reviewer_ChBY · 2022-07-26
**A benchmark for some visual-language models**

**Rating:** 6
**Confidence:** 4
**Correctness:** Yes
**Clarity:** Yes

**Strengths:**

1, V-L pretraining is very hot these days.
2, The benchmarking includes settings such as zero-shot, few-shot, and full-shot settings.
3, Reproducible Toolkit & Language-augmented Adaptation Methods.
4， A lot of OD datasets. Better than just using LVIS.



**Weaknesses:**

1, Some existing conclusions and findings in the original CLIP paper, e.g. CLIP is better than some self-supervised methods such as MoCov3, MAE; CLIP sample efficiency is better than supervised ViT. Not surprising.
2, Limited comparison in the Image classification parts (only  CLIP) in Industry Track. How about BLIP, FILIP, DECLIP? The authors should compare more methods to have a more complete benchmarking.
3, How about fully supervised results for object detection without language aids in Table 4. Not consistent with Table 3.
4. More comparisons can be added in Table 4 such as X-DETR and so on.
5. How about the automatic hyperparameter tuning toolkit? Do you have a comparison of the benefit of using it?

**Additional Feedback:**

None

**Documentation:**

URL not provided

**Relation To Prior Work:**

Yes

**Summary And Contributions:**

Introduce ELEVATER, a benchmark and toolkit for evaluating (pre-trained) language-augmented visual models.
(i) Datasets. As downstream evaluation suites, it consists of 20 image classification datasets and 35 object detection datasets, each of which is augmented with external knowledge.
(ii) Toolkit. An automatic hyper-parameter tuning toolkit is developed to facilitate model evaluation on downstream tasks.
(iii) Metrics. A variety of evaluation metrics are used to measure sample-efficiency (zero-shot and few-shot) and parameter-efficiency (linear probing and full model fine-tuning).

---

> ### Author Response · Authors · 2022-08-23
> **Response to Reviewer ChBY**
>
> > 1. Some existing conclusions and findings in the original CLIP paper, e.g. CLIP is better than some self-supervised methods such as MoCov3, MAE; CLIP sample efficiency is better than supervised ViT. Not surprising.
>
> A: In the original CLIP paper (Table 10), the authors indeed compare CLIP with self-supervised methods (including MoCo-v1 and MoCo-v2) and supervised methods (including ViT). **All of the methods are before Feb. 2021**. However, in the past 1.5 year, both self-supervised methods and supervised methods with Transformers have been advancing significantly. We revisit the comparisons, with following differences:
>
> - More advanced variants of self-supervised methods (including MoCov3 and MAE) and supervised methods (including DeiT) are studied, which are NOT covered in the original CLIP paper. The question of which learning paradigm (image-text learning, self-supervised learning, supervised learning) is the best for generic vision backbone training is a hot and important topic (often time debatable), the conclusions are evolving over time, and remain unclear to the community.
> - Our comparisons are more rigorous, for example in Table 10, ViT-B16 network architectures are used for all methods. In contrast, different Transformers and ConvNet architectures are used in the original CLIP paper.
> - Both linear probing and fine-tuning are considered in our comparisons, while only linear probing is considered in the original CLIP.
>
> ---
>
> > 2. Limited comparison in the Image classification parts (only CLIP) in Industry Track. How about BLIP, FILIP, DECLIP? The authors should compare more methods to have a more complete benchmarking.
>
>
> A:  To have a more complete benchmarking, we conduct more experiments and add a new table (Please see Table 14  Appendix  for detailed results).
>
> |Pre-trained Checkpoint |Backbone| Pre-trained Dataset | Zero-shot Settings (Averaged Score) |
> |-----|--------|-----|--------|
> | ***Academic Track***|
> |CLIP       | ViT-B32       | YFCC (15M) | 32.0 |
> |DeCLIP   | ViT-B32       | YFCC (15M) | 37.9 |
> |FILIP       | ViT-B32       | YFCC (15M) | 34.5 |
> |SLIP       | ViT-B32       | YFCC (15M)  | 31.2 |
> | ***Industry Track***|
> |CLIP       | ViT-B32       | WebImageText (400M) | 56.8 |
> |DeCLIP   | ViT-B32       | DeCLIP   (88M) | 51.0 |
> |OpenCLIP | ViT-B32       | LAION (400M) | 57.5 |
> |CLIP       | ViT-B16       | WebImageText (400M) | 60.0 |
> |OpenCLIP   | ViT-B16       | LAION (400M) | 59.1 |
> |CLIP       | ViT-L14       | WebImageText (400M) | 65.9 |
> |OpenCLIP       | ViT-L14       | LAION (400M) | 62.5 |
> |CLIP (px336)    | ViT-L14      | WebImageText (400M) | 68.8 |
>
> Note that we did not add BLIP to comparison as it uses ViT checkpoint pretrained on ImageNet22K (which includes ImageNet-1K) to initialize its visual encoder, which can create an unfair comparison with other approaches and is thus not used for comparison in either academic or industry track.
>
> Since this research topic is popular, and some best model checkpoints are private, we are also organizing an ECCV workshop to attract more checkpoints (either public or private) to contribute numbers. We strongly believe it is a collaborative community-effort to make the full benchmarking. Please check the update-to-date information on the leaderboard in the future  (https://computer-vision-in-the-wild.github.io/eccv-2022/).
>
> ---
>
> > 3. How about fully supervised results for object detection without language aids in Table 4. Not consistent with Table 3.
>
> A:Dyhead [*] is the fully supervised object detection model. It achieved near SoTA performance on COCO, based on paperwithcode leaderboard (https://paperswithcode.com/sota/object-detection-on-coco). It represents one of the best fully supervised object detection model.
>
> [*] Dynamic head: Unifying object detection heads with attentions, ICCV 2021
>
> ---
>
> > 4. More comparisons can be added in Table 4 such as X-DETR and so on.
>
> A: Unfortunately, the checkpoint and codebase of X-DETR is not publicly available, though we have reached out to the authors. We are running a workshop challenge based on the benchmark (https://computer-vision-in-the-wild.github.io/eccv-2022/), and are actively working together with other teams to onboard more methods for comparisons. Due to limited time, we are not able to provide more comparisons at this moment.
>
> ---
>
> > 5. How about the automatic hyperparameter tuning toolkit? Do you have a comparison of the benefit of using it?
>
> A: Our current automatic hyperparameter tuning toolkit is based on a simple strategy: grid-search. The baseline we could think of is manual tuning of the optimal hyper-parameters. Compared with this manual tuning, we believe automatic tuning makes it feasible to transfer models to a large number of downstream datasets in our benchmark. Please let us know if there is more doable baseline we could compare to reflect the benefit of our tuning toolkit.

---

### Official Review · Reviewer_S6nw · 2022-07-27

**Rating:** 7
**Confidence:** 3
**Correctness:** Yes
**Clarity:** The paper needs to be rewritten to be…

**Strengths:**

There is plenty of work done to merge the datasets and create automatic annotations. There are detailed baselines for all combination of training methods and there is also analysis of the results.

**Weaknesses:**

The paper is very hard to read. The structure of paper is very scattered and it is not easy to follow central topic of the paper. The reader is overwhelmed with the details, but as they are not explained it is hard to understand them. Also the language quality is very low that makes the paper almost unreadable.

**Additional Feedback:**

Please restructure the paper and polish the English.

**Documentation:**

The supplementary materials includes all the necessary parts

**Ethics:**

OK

**Relation To Prior Work:**

Yes

**Summary And Contributions:**

The paper presents dataset and benchmark tools for image classification and object detection. The authors combined 55 datasets and created benchmark and evaluation tools. They augmented datasets with Wordnet, Wiktionary and GPT3 language inputs and created baselines for the combined datasets.

---

> ### Author Response · Authors · 2022-08-23
> **Response to Reviewer S6nw**
>
> > The paper is very hard to read. The structure of paper is very scattered and it is not easy to follow central topic of the paper.
>
> A: The central topic of this paper is to provide a benchmark and toolkit to evaluate the task-level transfer ability of pre-trained visual models. We add a visual illustration of the pipeline in Figure 1 to describe the process of how a pre-trained checkpoint is evaluated, and add a paragraph to describe the structures & organization of the paper in Line 57-61. It is easy to see that the paper focuses on describing two most important modules in the pipeline: benchmark and toolkit.
>
> > The reader is overwhelmed with the details, but as they are not explained it is hard to understand them.
>
> A: Due to the limited space, we are not able to put all the details in the main paper (10 pages for main paper and 12 pages for appendix). Therefore, the main paper is presented with the assumption that the readers are familiar with techniques in generic vision backbone training, and the recent trend in language-image pre-trained models. Based on the clear and insightful requests/suggestions from all other reviewers, the clarity of the paper is improved from various aspects, including:
>
> - A pipeline is added in Figure 1
> - Prompt construction process with and without knowledge in Section C4 and referred in the main paper.
> - The notations about language-augmented model adaptation and in Figure 3 are revised.
>
> Reviewer S6nw may read our response to other reviews for the improvements, or provide more instructional comments so that we could address your concerns. For example, which specific places (including Line/Section/Table/Figure numbers) that Reviewer S6nw feels overwhelmed and requests more explanations.
>
> > Also the language quality is very low that makes the paper almost unreadable.
>
> A: We will proofread the draft with professionals & native speakers; Meanwhile, we would be happy to learn more specific suggestions.

---

> ### Author Response · Authors · 2022-08-28
> **Response to Reviewer S6nw (Reminder)**
>
> Dear Reviewer S6nw,
>
> During the past 3 days, we proofread our paper multiple times among different co-authors again, and believe that the language quality is improved. Major updates include:
>
> - Typos and grammar errors are corrected.
> - The description of ``Section 4 Toolkit'' is improved. The model adaptation/initialization methods are re-named, following the format of (random-init vs language-init) and (one-projection vs two-projection). Figure 3 & Table 3 are updated accordingly.
>
> We further edit the paper to include more detailed description, with the following structure adjustment:
> -  A new Section D on "Evaluation Metrics with Efficiency Considerations: Adaptation Cost" is added in Appendix. We discuss our design philosophy behind our evaluation metrics (*sample-efficiency* and *parameter-efficiency*), with a focus on model adaptation cost. A new Figure 8 is added to illustrate the 2D cost chart, which motivates our choice of two-dimensional efficiency considerations.
>
> -  A new Section G.1 on "A Taxonomy of Pre-trained Vision Models" is added in Appendix. We discuss our taxonomy hierarchy of existing pre-trained vision models, from the perspectives whether language and / or knowledge is employed in pre-training. A new Table 10 is added to illustrate the taxonomy hierarchy, with pre-trained checkpoints used in this paper as examples. From this table, one may immediately see *language-augmented vs language-free* and *knowledge-augmented vs knowledge-free* pre-trained models. More background on knowledge-augmented pre-trained models is also added in this section.
>
> Please let us know any of your specific points to improve the paper. We are happy to revise it accordingly.

---

### Official Review · Reviewer_KfmV · 2022-07-28
**New benchmark for task-level transfer of visual models pretrained with natural language supervision.**

**Rating:** 7
**Confidence:** 3
**Correctness:** Everything can be checked publicly an…
**Clarity:** See weaknesses

**Strengths:**

The benchmark is useful to evaluate task-level transfer. It contains many datasets with two task types (IC and OD) instead of only one. It should help achieve reproducibility in this field. Adding knowledgebases to the datasets is very helpful to standardize evaluation there, as otherwise every work would build its own knowledgebase.

In the general the work is exhaustive in all dimensions, there are enough tested models, ablated evaluation strategies, metrics to give a complete picture of both the quality of the benchmark as well as the performance of the models tested on that benchmark.

The toolkit code and the datasets are all released publicly so no information is missing to the reader.

**Weaknesses:**

This work has a lot of content with results distributed over various tables and figures in both the paper and supplementary, so it can be hard to understand. To elaborate:

In table 2 you list that you evaluated 7 IC and 4 OD, models but some of the results seem to be missing, e.g. I could not find the K-LITE performance for OD. Maybe you could create a table in the appendix in the style of table 2 where you list the results for each model so the reader doesn't have to collect this information from various places.

It seems you have improved over the original CLIP performance (line 191 "superior empirical performance") with the new language initialization method. This is hard to confirm with so many tables, methods, evaluation strategies and so on and also since you used only a small CLIP (B32). What would be interesting would be to compare with the original CLIP paper numbers directly, e.g. in their paper appendix table 10 row 8 model "L/14-336px" on ImageNet with linear probe achieves 85.4% accuracy, can you improve over this with your new language init methods?

- Figure 2 has some weaknesses that make it hard to understand:
    - |B| is not defined anywhere, this is probably the batchsize / dataset size. If it is not necessary it could be removed, i.e. having the figure and text talk about a single image x instead of many images. Otherwise it should be defined.
    - Line 170 describes projection layers W_u for image and W_v for text, however in figure 2b) its called W_t instead (probably a typo).
    - In figure 2b) the output of text encoder is not shown, as I understand it's shape DxK (class text prompts encoded as text), this should be added explicitly.
    - Having W with shape DxK in figure 2a), 2e) but with shape PxK in 2c) is confusing, if the shape changes it should probably get a different name.
    - Text uses lowercase u, v and figure uses uppercase U, V, without explicitly defining the difference. It can be assumed that lowercase are single datapoints and U, V are multiple datapoints, but this should be made explicit.
    - If I understand it correctly, methods 2b) and 2d) end up being mathematically the same during zero-shot but different during finetuning. If this is true, this information should be added to chapter 4 instead of having the reader deduct it.

Table 3: You are comparing the methods in figure 2c,d,e (Random, Lang-S, Lang-M). Why not compare also the original CLIP setting 2b)? Is the comparison unfair, if yes, why?

Table 3: Why is no result given for row 2 (CLIP, LP, Language-S)? I understand that with random linear heads zeroshot doesn't work, but with Language-Init (Separate) as in Fig 2d) it should work.

In line 239 the claim "CLIP outperforms Supervised ViT consistently" is not clear from table 3 as the Supervised ViT FT matches CLIP performance with enough data (50-shot and full). This contrasts your finding in line 48: "Leveraging ... text and vision ... consistently yields better performance" so you should try to explain why here this is not true and vision-only matches the performance of vision-text instead of vision-text yielding better performance.

The prompts are not described at all, this could be done in the appendix. e.g. are the prompts actually sentences like "This is an image of a dog." or only a word like "dog".

It is not stated exactly how the external knowledge is added, the reader has to infer from the hints in line 191 and 292 that the knowledge is added to the class prompt before feeding everything into the text encoder f. Details are missing, e.g. with GPT-3 there are 5 knowledge sentences generated, do you concatenate all 5 to the prompt, or generate multiple prompts, or etc. Also, which special tokens (CLS / SEP) or similar are used when constructing the input to the text encoder?

The hyperparameter tuning is very simple with grid search and only the 2 hyperparemeters learning rate and weight decay. Why was this approach chosen? Why not a more sophisticated method like BOHB (2018), DEHB (2021) or others?


**Additional Feedback:**

No

**Documentation:**

Yes

**Relation To Prior Work:**

Yes

**Summary And Contributions:**

The authors propose a new benchmark to evaluate transferability of models that learn visual representations from language supervision consisting of 20 image classification (IC) and 35 object detection (OC) datasets. They include an evaluation toolkit with a simple hyperparameter optimization method and define metrics for sample-efficiency and parameter-efficiency. The benchmark additionally includes a collected external knowledgebase for each dataset. The overall goal is the evaluate task-level transfer across domains, the ability of pretrained models to adapt to new tasks.

The authors evaluate 7 IC / 4 OD pretrained models and propose new model adaption methods, i.e. new methods to evaluate a model like CLIP. Extensive evaluation of the models on the new benchmark is done with regards to usefulness of language pretraining, choice of language-augmented initialization method, sample efficiency, parameter efficiency, external knowledge.

---

> ### Author Response · Authors · 2022-08-23
> **Response to Reviewer KfmV (Part 1)**
>
> > In table 2 you list that you evaluated 7 IC and 4 OD, models but some of the results seem to be missing, e.g. I could not find the K-LITE performance for OD. Maybe you could create a table in the appendix in the style of table 2 where you list the results for each model so the reader doesn't have to collect this information from various places.
>
> A: In the paper, we present isolated tables to present the most relevant information to reflect each individual point. Thanks for the suggestion to provide full results (so the reader doesn't have to collect this information from various places). To address this, we provides the followings:
> - We create excel files to include the full results of IC and OD. The are also submitted as a part of the supplementary material, and we will also provide the links for reader to navigate from our benchmark website:
> [[IC results]](https://docs.google.com/spreadsheets/d/10qoHoxi_MMlGuncg_crfaT5zJ19ystEWpdkwsH9HjJI/edit?usp=sharing)
> [[OD results]](https://docs.google.com/spreadsheets/d/1vPiIJH_HSuQCKO4LEO4fyx6Ea0m9Ojbrg6rtS0BuyiU/edit?usp=sharing)
>
> - We also suggest readers to check out the leaderboard of each Challenge/Phase for up-to-date information in the future (https://computer-vision-in-the-wild.github.io/eccv-2022/).
>
> Note that only the zero-shot OD performance for KLITE is provided, as the performance gap between KILITE and GLIP-A is getting closed with more shots.
>
>
> ----
>
> >  It seems you have improved over the original CLIP performance (line 191 "superior empirical performance") with the new language initialization method. This is hard to confirm with so many tables, methods, evaluation strategies and so on and also since you used only a small CLIP (B32). What would be interesting would be to compare with the original CLIP paper numbers directly, e.g. in their paper appendix table 10 row 8 model "L/14-336px" on ImageNet with linear probe achieves 85.4% accuracy, can you improve over this with your new language init methods?
>
> A: The proposed language-initialization method significantly improves the random-initialization baseline **in the few-shot settings**. The advantage of language-initialization method is becoming less important when more training data is involved, for example, the performance of the two methods are similar in the full-shot settings. This conclusion is shown in Table 3. Therefore, we hypothesize the proposed  language-initialization method can hardly improve the full-shot ImageNet linear probing performance.
>
> To further demonstrate the effectiveness of the proposed language-initialization method in few-shot settings, we conducted several other experiments, summarized in a new table below (See details in Table 13  in Appendix).
>
> | Pre-trained Checkpoint | Backbone | Adaptation   | Language-initialization |   Random-initialization    | Improvement |
> | :---        |    :----:   |    :----:   |    :----:   |     :----:   |           ---: |
> |      *Base Models (Fine-tuning)*        |
> | DeCLIP | ViT-B32 |  Fine-tuning  |  58.82 | 64.51 |   **+5.69** |
> | OpenCLIP | ViT-B32 |  Fine-tuning  |  34.59  | 64.83   |  **+30.24**  |
> | OpenCLIP | ViT-B16 |  Fine-tuning  |  27.64 | 66.24 |   **+38.60** |
> |      *Base Models (Linear probing)*        |
> | DeCLIP | ViT-B32 |  Linear probing  |  57.25  | 62.45  |   **+5.20** |
> | OpenCLIP | ViT-B32 |  Linear probing  | 61.92  | 68.64     |  **+6.72**  |
> | OpenCLIP | ViT-B16 |  Linear probing  |  62.88  | 69.73   |   **+6.85** |
> |      *Large Models (Linear probing)*        |
> | OpenCLIP | ViT-L14 |  Linear probing  |  66.50 (1.64)  | 72.52 (1.27)   |  **+6.02** |
> | **CLIP** |  **ViT-L14 (336px)** |  Linear probing  | 68.35 | 75.20  |  **+6.85** |
>
> The experiments are conducted in three dimensions: different backbone, different pre-training methods, different adaptation approach (linear probing / fine-tuning).
> - With the same ViT-B32 backbone, language initialization improves the checkpoints pretrained by CLIP, DeCLIP and OpenCLIP consistently by ~6% average score.
> - With the same pre-training (OpenCLIP), language initialization improves checkpoints of different backbones (ViT-B32/B16/L14) consistently by ~6% average score.
> - With the largest CLIP-L14-336px checkpoint, it also improves significantly by 6.85% average score.
> - With fine-tuning as the adaptation strategy, language initialization improves over OpenCLIP checkpoints with ViT-B32/B16 backbones significantly and consistently by 30.2% and 38.6% average score.

---

> ### Author Response · Authors · 2022-08-23
> **Response to Reviewer KfmV (Part 2)**
>
> >  Figure 2 has some weaknesses that make it hard to understand:
>
> A: Thanks for your careful read. We have updated the paper to address these issues accordingly, and summarize our edits as follows:
>
> > $|\mathcal{B}|$ is not defined anywhere, this is probably the batchsize / dataset size. If it is not necessary it could be removed, i.e. having the figure and text talk about a single image x instead of many images. Otherwise it should be defined.
>
> A: $|\mathcal{B}|$ is batch size
>
> > Line 170 describes projection layers W_u for image and W_v for text, however in figure 2b) its called W_t instead (probably a typo).
>
> A: We remove $W_u$. Now it is consistent that $W_v$  for image and $W_t$ for text
>
> In figure 2b) the output of text encoder is not shown, as I understand it's shape DxK (class text prompts encoded as text), this should be added explicitly.
>
> A: We revise the figure to add: $H_t \in \mathbb{R}^{D \times K}$
>
> > Having W with shape DxK in figure 2a), 2e) but with shape PxK in 2c) is confusing, if the shape changes it should probably get a different name.
>
> A: We revise the figure to add $W_m$
>
> >Text uses lowercase u, v and figure uses uppercase U, V, without explicitly defining the difference. It can be assumed that lowercase are single datapoints and U, V are multiple datapoints, but this should be made explicit.
>
> A: Clarification is added in Line 186-187 ``*Note that lowercase $u$ and $v$ are single feature vectors while $U$ and $V$ are a batch with multiple feature vectors.*''
>
> > If I understand it correctly, methods 2b) and 2d) end up being mathematically the same during zero-shot but different during finetuning. If this is true, this information should be added to chapter 4 instead of having the reader deduct it.
>
> A: Thanks for your in-depth understanding of this part. They are *largely* equivalent (perhaps in terms of performance evaluation), but not exactly the same mathematically. We address your concern in three aspects:
>
> -  This equivalence information is explicitly added in Line 199-201: ``*... shown in Figure 3 (d). In this way, the visual and text heads are separated. Note that the Separate-Head scheme is also largely equivalent to Figure 3 (b) in zero-shot settings. Please see Appendix Section E for discussions*''
>
> - In the paper, we have mentioned equivalence in Line 223:``*With the proposed language-init method, one can ensure that few-shot performance is always better than zero-shot, as we essentially reduce to zero-shot when zero iteration is updated in our language-init method.*''
>
> - Appendix Section E is dedicated to discuss the equivalence. The reason that they are not exactly the same is due to the implementation details in literature, including temperature and normalizations. This discussion is referred in Line 201 in the main paper.
>
>
> ---
>
> > Table 3: You are comparing the methods in figure 2c,d,e (Random, Lang-S, Lang-M). Why not compare also the original CLIP setting 2b)? Is the comparison unfair, if yes, why?
>
> Table 3: Why is no result given for row 2 (CLIP, LP, Language-S)? I understand that with random linear heads zeroshot doesn't work, but with Language-Init (Separate) as in Fig 2d) it should work.
>
> A: Sorry for the confusion. We clarify that only one zero-shot number is reported for each pre-training checkpoint. Various model adaptation/init methods (LP or FT, Random, Lang-S, Lang-M) are independent from the zero-shot number, which is simply evaluated with the zero-shot evaluation method in Figure 2b. A table footnote is added to clarify this.
>
> ---
>
> > In line 239 the claim "CLIP outperforms Supervised ViT consistently" is not clear from table 3 as the Supervised ViT FT matches CLIP performance with enough data (50-shot and full). This contrasts your finding in line 48: "Leveraging ... text and vision ... consistently yields better performance" so you should try to explain why here this is not true and vision-only matches the performance of vision-text instead of vision-text yielding better performance.
>
> A: We revise the statement accordingly:
> - Line 260: ``*We see that the language-augmented model (CLIP) outperforms language-free model (Supervised ViT) for the limited data settings. The gap is closed when more training examples are observed (eg, 50-shot and full-shot). This is probably because the pre-training power is gradually dominated by larger-scale downstream training.*''
> - Line 48 ``*Leveraging both text and vision in these models consistently yields better performance than vision-only in few-shot settings*''

---

> ### Author Response · Authors · 2022-08-23
> **Response to Reviewer KfmV (Part 3)**
>
> > The prompts are not described at all, this could be done in the appendix. e.g. are the prompts actually sentences like "This is an image of a dog." or only a word like "dog". It is not stated exactly how the external knowledge is added, the reader has to infer from the hints in line 191 and 292 that the knowledge is added to the class prompt before feeding everything into the text encoder f. Details are missing, e.g. with GPT-3 there are 5 knowledge sentences generated, do you concatenate all 5 to the prompt, or generate multiple prompts, or etc. Also, which special tokens (CLS / SEP) or similar are used when constructing the input to the text encoder?
>
> A: We add the details of how to construct prompts with and without external knowledge as follows:
>
> - All the prompt templates are data-specific. They are summarized in one file in our toolkits [[vision_benchmark/datasets/prompts.py]](https://github.com/Computer-Vision-in-the-Wild/Elevater_Toolkit_IC/blob/main/vision_benchmark/datasets/prompts.py). Due to limited space, we decide to provide the link in our paper, instead of copying them into the Appendix.
>
> - To address the details of prompting with and without knowledge, we add Section C4 in Appendix named “Prompting and Knowledge". It provide the step-to-step guidance to illustrate prompting procedure.
>
> - We provide one example in Table 7, to illustrate prompt templates, knowledge, and prompt construction with and without external knowledge for the concept *pink primrose* in dataset *Oxford Flowers 102*.
>
> - The description to construct prompts with multiple knowledge items are added in Line 203-210 in the main paper.
>
> ---
>
> > The hyperparameter tuning is very simple with grid search and only the 2 hyperparemeters learning rate and weight decay. Why was this approach chosen? Why not a more sophisticated method like BOHB (2018), DEHB (2021) or others?
>
> A: We follow the original CLIP paper to implement a simple grid-search style tuning pipeline, and leave more sophisticated methods like BOHB and DEHB  as future work. This is now clarified in Line 168-169.

---

> > ### Comment · Reviewer_KfmV · 2022-08-26
> > **Thanks**
> >
> > Thank you for your detailed answers, I will happily keep the response at 7 (Good paper, accept).

---

### Author Response · Authors · 2022-08-23
**Summary of Response**

We sincerely thank all the reviewers for their time and their thoughtful comments and questions. We are encouraged that the reviewers find that:

- Our work is comprehensive (KfmV, S6nw, ChBY,  Jeow  onGS) in terms of benchmark construction and evaluation results, well motivated in its timing contribution to a popular topic (KfmV, ChBY) and the use of external knowledge (KfmV, Jeow)
- The toolkit makes research reproducible (KfmV, ChBY), and the proposed language-init method is novel and effective (Jeow, onGS).
- It is considerate to make both academic and industry tracks in order to increase inclusivity to all levels of accessibility (Jeow, onGS).
- The paper is well written (Jeow, onGS)

We attempted our best to address the questions as time allowed. We believe the comments & revisions have made the paper stronger and thank all the reviewers for their help. Please find individual responses to your questions below. More response with quantitive results will be provided once available.


---

Some major updates of the paper include:
- Details of prompt templates, prompt construction with and without external knowledge
- Pipeline is added in Figure 1
- 10+ additional model checkpoints are evaluated for the effectiveness of language-initialization method, more baseline for Industry and Academic Tracks.
- Related works are added for discussions, including more OD datasets and benchmarks for language-images models.

---

### Meta-Review · Area_Chair_gAXy · 2022-09-08

**Recommendation:** Accept
**Confidence:** 4

**Metareview:**

The paper presents a new exciting benchmark for evaluating pre-trained vision+language models. It was reviewed by five experts, who are uniformly supportive.

---

### Decision · Program_Chairs · 2022-09-16

Accept